# IKL: Boosting Long-Tail Recognition with Implicit Knowledge Learning

## Abstract

In the field of visual long-tailed recognition, the long-tailed distribution of image representations often raises two key challenges: (1) the training process shows great uncertainty (e.g., uncertainty in the prediction of augmented views by the same expert for the same sample) and (2) a marked bias in the model's prediction towards the head class. To tackle the above issue, we propose a novel method termed Implicit Knowledge Learning (IKL) to extract the knowledge hidden in long-tail learning processes, aiming to significantly improve performance in long-tail recognition. Our IKL contains two core components: Implicit Uncertainty Regularization (IUR) and Implicit Correlation Labeling (ICL). The former method, IUR, exploits the uncertainty of the predictions over adjacent epochs. Then, it transfers the correct knowledge to reduce uncertainty and improve long-tail recognition accuracy. The latter approach, ICL, endeavors to reduce the bias introduced by one-hot labels by exploring the implicit knowledge in the model: inter-class similarity information. Our approach is lightweight enough to plug-and-play with existing long-tail learning methods, achieving state-of-the-art performance in popular long-tail benchmarks. The experimental results highlight the great potential of implicit knowledge learning in dealing with long-tail recognition. Our code will be open sourced upon acceptance.

## 1 Introduction

Real-world scenarios often exhibit a long-tail distribution across semantic categories, with a small number of categories containing a large number of instances, while most categories have only a few instances Zhang et al. (2021b). Dealing with Long-Tail Recognition (LTR) is a challenge as it involves not only addressing multiple small-data learning problems in rare classes but also handling highly imbalanced classification across all classes. In addition, the inherent bias towards the high-frequency (head) classes may cause the low-frequency (tail) classes to be neglected, leading to inaccurate classification results.

To address this challenge, many approaches have explored LTR in order to learn well-performing models from long-tailed datasets, such as data re-sampling Buda et al. (2018); Byrd & Lipton (2019), re-weighting Khan et al. (2017); Lin et al. (2018); Cui et al. (2019); Cao et al. (2019); Xie & Yuille (2019); Menon et al. (2020b); Alshammari et al. (2022), decoupling learning Kang et al. (2019); Xu et al. (2022), contrastive learning Yang & Xu (2020); Kang et al. (2020); Wang et al. (2021b); Zhu et al. (2022); Cui et al. (2021), Calibration Zhong et al. (2021), transfer learning Parisot et al. (2022), and multi-expert ensemble learning Xiang et al. (2020); Wang et al. (2020); Cai et al. (2021); Zhang et al. (2022); Li et al. (2022); Zhao et al. (2023).

In this paper, we propose a novel method named Implicit Knowledge Learning (IKL), aimed at exploring the hidden knowledge within the long-tail learning process to significantly improve performance in long-tail recognition tasks. Previous long-tail learning methods Li et al. (2022) have unveiled the prediction uncertainty encountered in such tasks: An expert often exhibits uncertainty when predicting different augmentation views of the same sample; similarly, two experts in the same experimental settings also exhibit such uncertainty when predicting the same augmented view. The method NCLLi et al. (2022) explicitly creates a variety of augmented samples, employing collaborative learning to grasp the consistent and richer information among these augmentations. Besides, methods LEMF Xiang et al. (2020), RIDE Wang et al. (2020), and SADE Zhang et al. (2022) reduce the model

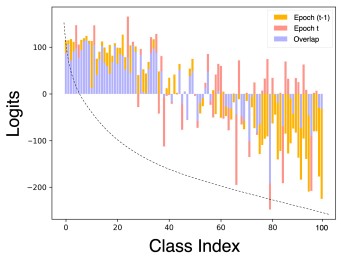 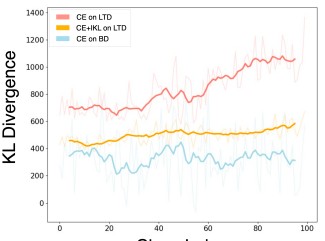 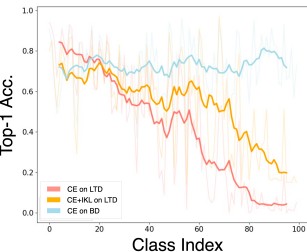

(a) Logits between the same networks that across random adjacent epochs.

(b) Comparison of KL Divergence Across Classes for Different Methods and Data Distributions.

(c) Comparison of Top-1 accuracy Across Classes for Different Methods and Data Distributions.

Figure 1: The comparisons of model outputs (logits) and Kullback–Leibler (KL) distance. The analysis is conducted on CIFAR100-LT dataset with an Imbalanced Factor (IF) of 100. The logits, KL distance, and accuracy are visualized on the basis of the whole test set and then the average results of each category are counted and reported. **(a):** The dashed line represents the direction of the long-tail distribution in data volume, and the prediction consistency (Overlap) of the head class is significantly higher than that of the tail class. **(b) and (c):** The figure compares the per-class KL-Divergence and top-1 accuracy results of Cross-Entropy (CE) on Long-Tail Data (LTD) and Balanced Data (BD), as well as the results on LTD after incorporating our proposed method (IKL). Compared to the original Cross-Entropy, our method (IKL) not only significantly reduces the overall prediction divergence but also alleviates the divergence imbalance caused by the inconsistency in predictions between head and tail classes. Concurrently, our method significantly enhances the model's accuracy on the test set and mitigates the phenomenon where the head class accuracy substantially surpasses that of the tail class due to data imbalance.

uncertainty by the ensemble model, achieving improved long-tail recognition performance. However, an unexplored implicit uncertainty persists in existing long-tail learning approaches: in adjacent training epochs, there are also different augmentation views of the same sample, and the prediction of the same expert also has uncertainty. To analyze this, we visually depict the relationship between model predictions (logits) across a random adjacent epoch in Figure 1. As illustrated in Figure 1 (a), the model exhibits lesser overlap in the tail class compared to the head class. Concurrently, as shown in Figure 1 (b), the KL divergence between predictions across adjacent epochs is larger for the tail class. These observations indicate that the uncertainty in predictions for the tail class across adjacent epochs is relatively more significant compared to the head class.

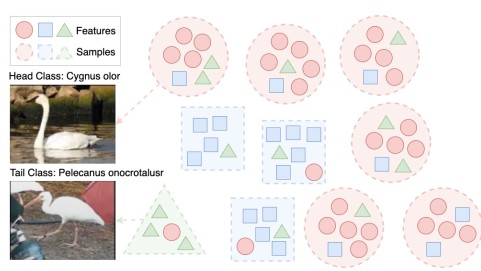

Figure 2: Correlation of features among different samples in long-tailed data.

Another kind of implicit knowledge is in the features of the model that can help us reconstruct the labeling space. Traditional one-hot labels, due to their inability to carry inter-class similarity information, exacerbate the model's bias towards the head class in long-tail learning. As demonstrated in Figure 2, when head class "Cygnus olor" shares some similar features with tail class "Pelecanus onocrotalus", one-hot labels during the supervision process strictly categorize all these features under head class "Cygnus olor". Given the large sample size of the head class in the long-tailed dataset, this type of supervision can mislead the model to misclassify the tail class "Pelecanus onocrotalus" as the head class "Cygnus olor", thus exacerbating the model's recognition bias of the head class. Although previous efforts to refine label space, such as label smoothing Szegedy et al. (2016), Mixup labelZhang et al. (2017); Chou et al. (2020), and label-aware smoothing Zhong et al. (2021), have alleviated the bias towards the head class to some extent, they overlook the intrinsic inter-class similarity information in the model during the label space reconstruction process.

To address the above issues, we propose Implicit Knowledge Learning (IKL), which contains two core components: Implicit Uncertainty Regularization (IUR) and Implicit Correlation Labeling

(ICL). Our IUR utilizes KL divergence to learn the uncertainty in the output distribution of correct predictions across adjacent epochs. As depicted in Figure 1 (b), our method minimizes the uncertainty between prediction distributions and achieves markedly improved test accuracy. On the other hand, our ICL seeks to reduce the bias introduced by one-hot labels by constructing class centers during the training process. Then it computes a cosine similarity matrix to reconstruct the label space. Due to the lightweight design of these two methods, our approach could easily, as a plug-to-play method, integrate with existing long-tail learning methods and boost them to achieve state-of-the-art performance. These results underscore the efficacy and potential of IKL in addressing the challenges faced in long-tail recognition tasks.

## 2  RELATED WORK

**Re-sampling/weighting methods** Re-sampling methods, including over-sampling minority classes or under-sampling majority classes, have been proposed to balance the skewed data distribution. Over-sampling minority classes by duplicating samples can lead to over-fitting Buda et al. (2018). On the other hand, under-sampling majority classes may result in loss of crucial information and impaired generalization ability Japkowicz & Stephen (2002). Re-weighting methods assign different weights to different classes based on loss modification or logits adjustment Khan et al. (2017); Lin et al. (2018); Cui et al. (2019); Cao et al. (2019); Xie & Yuille (2019); Aimar et al. (2023); Menon et al. (2020b); Liu et al. (2019); Wang et al. (2021a). However, these methods can potentially hurt representation learning, and it has been observed that decoupling the representation from the classifier can lead to better features Malach & Shalev-Shwartz (2017); Zhou et al. (2020).

**Ensemble-based methods.** These methods used multiple experts with aggregation methods to reduce the uncertainty, and are receiving more and more attention due to their effectiveness on long-tailed recognition. LFME Xiang et al. (2020) trained different experts with different parts of the dataset and distills the knowledge from these experts to a student model. RIDE Wang et al. (2020) optimized experts jointly with distribution-aware diversity loss and trains a router to handle hard samples. SADE Zhang et al. (2021a) proposed test-time experts aggregating method to handle unknown test class distributions.

**Label space adjustment** There has also been some previous work on adjusting the labelling space to prevent the model from over-fitting the header class, such as the universal method label smoothing Szegedy et al. (2016), Mixup Zhang et al. (2017). Recently, long-tail methods Chou et al. (2020); Zhong et al. (2021) consider the category frequencies in the reconstruction gave better results. However, these methods do not consider inter-class similarity information, and this knowledge is necessary when working with existing long-tail methods, which our method explores.

**Knowledge Distillation based methods.** HintonHinton et al. (2015) first proposed the concept of knowledge distillation. Knowledge distillation has gradually evolved from an offline process Peng et al. (2019); Hinton et al. (2015); Passalis & Tefas (2018) to an online process Chen et al. (2020); Guo et al. (2020); Zhang et al. (2018), and Self-distillation Zhang et al. (2019); Kim et al. (2021). For long-tailed recognition, knowledge distillation is always used to balance the predictions of head and tail classes Xiang et al. (2020); He et al. (2021); Parisot et al. (2022); Li et al. (2022); Park et al. (2023). Parisot et al. (2022) transfers knowledge of features from the head class to the tail class in classifier, but not guaranteeing the correctness of features. NCLLi et al. (2022) proposed a nested balanced online distillation method to collaboratively transfer the knowledge between any two expert models and explicitly augmented samples. While previous knowledge distillation strategies under explore the uncertainty in adjacent epochs. Our method discovers the uncertainty information between epochs and eliminates it effectively, bringing about an improvement in long-tail recognition results.

## 3  METHOD

In this section, we propose a new method called Implicit Knowledge Learning (IKL) to regularize the inconsistency of predictions. The framework is shown in Figure. 3, which includes two components: Implicit Uncertainty regularization (IUR) and Implicit Correlation Label (ICL). In the following part, we introduce the components in detail.

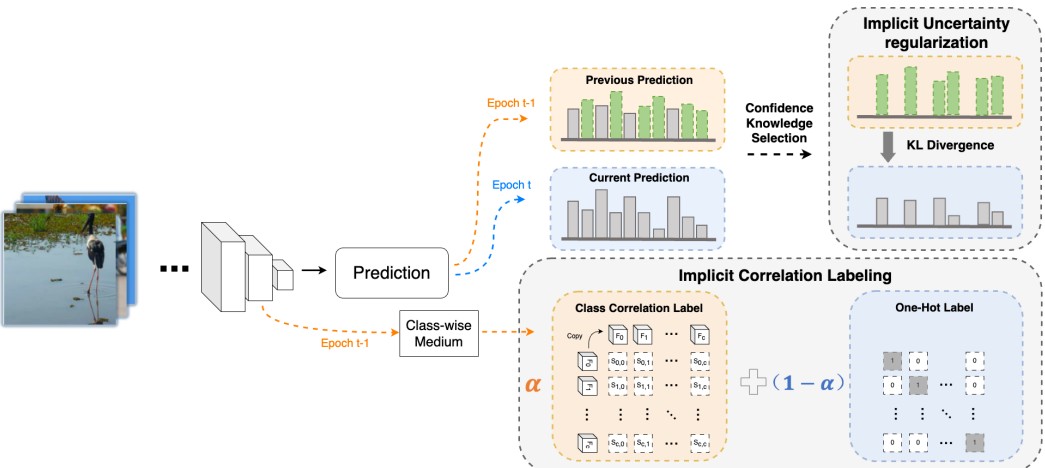

Figure 3: The framework of our method. The prediction of the previous epoch (t-1) serves as a soft label to regularize the prediction of the current epoch (t). During the regularization process, we first use Confidence Knowledge Selection (CKS) to filter out correctly predicted samples (indicated in green). Then, we employ Implicit Uncertainty Regularization (IUR) to regularize the uncertainty. Meanwhile, we compute the median of the features from the previous epoch to represent the characteristic features. Then the inter-class feature correlations are characterized using cosine similarity, resulting in a similarity matrix that serves as soft class-correlation labels for each category. By integrating these soft labels with one-hot labels in a weighted manner, we derive the ultimate supervisory labels for the model's learning process, a method we term Implicit Correlation Labeling (ICL).

## 3.1 PRELIMINARIES

Long-tailed recognition involves the learning of a well-performance classification model from a training dataset that is characterized by having a long-tailed category distribution. For a clear notation, we write a C-classes labeled dataset as $\mathbb{D} = \{(x_i, y_i)|1 \leq i \leq n\}$, which $x_i$ is the $i$-th training sample and $y_i = \{1, ..., C\}$ is its ground-truth label. In this context, we use $n_j$ to represent the number of training samples for class $j$, while $N = \sum_{j=1}^{C} n_j$ denotes the total number of training samples. To simplify our discussion, we assume that the classes are arranged in decreasing order, such that if $i < j$, then $n_i \geq n_j$. Furthermore, an imbalanced dataset is characterized by a significant disparity in the number of instances between different classes, with some classes having significantly more samples than others, i.e., $n_i \gg n_j$.

Suppose that a Softmax classifier is used to model a posterior predictive distribution, i.e., given the input $x_i$, the predictive distribution is:

$$p_i(x_i; \Theta) = \frac{e^{(v_i^k/\tau)}}{\sum_c e^{(v_i^c/\tau)}}, \tag{1}$$

where $v_i = \{f(x_i; \Theta), W\}$ denotes the logits of DNNs for instances $x_i$ which are calculated by feature $f(x_i; \Theta)$ and classifier weight $W$, and $\tau > 1$ is the temperature scaling parameter (a higher $\tau$ produces a "soften" probability distribution Hinton et al. (2015)).

## 3.2 IMPLICIT UNCERTAINTY REGULARIZATION (IUR)

The goal of IUR is to explore implicit uncertainty knowledge during long-tailed training to enhance recognition. From the above analysis 1, we learn that there is currently an implicit uncertainty knowledge in long-tail learning, i.e., the same expert has uncertainty about different augmentations for the same sample in adjacent epochs. To reduce the uncertainty of prediction between adjacent epochs, we employ KL divergence of the previous and current epoch's prediction distribution as the minimization object function. As demonstrated in Figure 3, at every epoch, our IUR optimizes the

current prediction to be closer to the previous prediction to transfer the useful and richer knowledge to reduce the uncertainty. We formulate the IUR denoted as:

$$\mathcal{L}_{IUR} = \sum_{x_i \in \mathbb{D}} KL(p_{i,t-1}(x_i; \Theta_{t-1}) || p_{i,t}(x_i; \Theta_t)) \tag{2}$$

In detail, our IUR employs the KL divergence function to perform optimization following soft distillationHinton et al. (2015) for instances, which can be formulated as:

$$KL(p_{i,t-1} || p_{i,t}) = \tau^2 \sum_{i=1}^{n} p_{i,t-1}(x_i; \Theta_{t-1}) log \frac{p_{i,t-1}(x_i; \Theta_{t-1})}{p_{i,t}(x_i; \Theta_t)}. \tag{3}$$

However, blindly transferring knowledge of neighboring predictions does not yield satisfactory results. For example, if the model misses the ground truth prediction for instance $x$, then the wrong knowledge is not suitable be transferred. Therefore, we employ a trick called **Confidence Knowledge Selection (CKS)** to prevent our method from introducing wrong knowledge, we only transfer the knowledge that is correctly classified. This method is a general variant of consistency learning employed in semi-supervised learning Sohn et al. (2020), and also very useful in our strategy. We define correctly classified instances set contain all correctly classified instances as:

$$\mathbb{D}_{CKS} = \{x_i \in \mathbb{D} | argmax(p_i(x_i; \Theta)) == y_i\}, \tag{4}$$

where $y_i$ denotes the ground-truth label of instance $x_i$. With the correct predictions of the previous epoch (t-1), we re-write the IUR with CKS as:

$$\mathcal{L}_{IUR} = \frac{1}{\left\|\mathbb{D}_{CKS}^{t-1}\right\|} \sum_{x_i \in \mathbb{D}_{CKS}^{t-1}} KL(p_{i,t-1}(x_i; \Theta_{t-1}) || p_{i,t}(x_i; \Theta_t)) \tag{5}$$

### 3.3 IMPLICIT CORRELATION LABELING (ICL)

In this section, we work on reconstructing the labeling space by looking for correlations of category features implicit in the model. Decoupled long-tail learning Malach & Shalev-Shwartz (2017) suggests that the bias of the long tail stems mainly from the classifiers rather than the backbone and the cosine distances leads to more unbiased feature boundaries. So, the features extracted by the backbone are less biased and cosine similarity is a good choice to learn relationships between features under long-tail distribution. Further to this, for C-th class we calculate the class center of $f_c$ by the medium of all features across the C-th class, which is denoted as:

$$f_c = medium_{x_i \in \mathbb{D}}(f(x_i; \Theta_{t-1})) \tag{6}$$

which medium is a function that calculates the median of the features for category $C$. We use the median rather than the mean to avoid outliers of the features produced by data augmentation. Then, we calculate the Correlation feature label by cosine similarity and reconstruct the label $\hat{y}$:

$$M = \frac{f \cdot f^T}{||f|| \cdot ||f||}, \hat{y} = Y + M \tag{7}$$

where Y is the label y after extending to label matrix. Finally, the ICL loss denoted as:

$$\mathcal{L}_{ICL} = \frac{1}{||\mathbb{D}||} \sum_{x_i \in \mathbb{D}} CrossEntropy(p(x_i; \Theta_t), \hat{y}) \tag{8}$$

## 3.4 IMPLEMENTATION.

During the training process, our proposed IKL can easily combine with the existing LTR methods. Therefore, the overall loss for implementation consists of two parts, the existing $\mathcal{L}_{LTR}$ loss for long-tailed recognition and our $\mathcal{L}_{IKL}$ for epistemic consistency. It is expressed as:

$$\mathcal{L} = (1 - \alpha)\mathcal{L}_{LTR} + \alpha(\mathcal{L}_{IUR} + \mathcal{L}_{ICL}) \tag{9}$$

where $\alpha$ is a linear parameter to trade-off the weight of $\mathcal{L}_{IUR}, \mathcal{L}_{ICL}$ and $\mathcal{L}_{LTR}$ with values in the range $0, 1$. In Sec. 5, we conduct experiments for the effect of the parameter $alpha$.

## 4 EXPERIMENTS

We present the experimental results on five widely used datasets in long-tailed recognition, including CIFAR-100/10-LT Krizhevsky et al. (2009), ImageNet-LT Liu et al. (2019), Places-LTLiu et al. (2019), and iNaturalist 2018Horn et al. (2017). Moreover, we undertake ablation studies specifically on CIFAR-100-LT datasets to gain deeper insights into the performance of our approach.

### 4.1 IMPLEMENTATION DETAILS.

**Evaluation Setup.** For classification tasks, after training on the long-tailed dataset, we evaluate the models on the corresponding balanced test/validation dataset and report top-1 accuracy. We also report accuracy on three splits of the set of classes: Many-shot (more than 100 images), Medium-shot (20 to 100 images), and Few-shot (less than 20 images).

**Method Implementation.** For IKL, incorporating the predictions from the model at the (t-1)-th epoch during training at the t-th epoch is essential. Two methods can be employed to obtain these past predictions. The first approach involves loading the model from the (t-1)-th epoch into memory at the start of the t-th epoch. This ensures that the past predictions for softening targets are computed during the forward passes. Alternatively, the second approach involves pre-saving the past predictions on disk during the (t-1)-th epoch and retrieving this information to compute the soft targets at the t-th epoch. Each approach has its advantages and drawbacks. The former method may require more GPU memory, while the latter method eliminates the need for additional GPU memory but necessitates more storage space to store past predictions. Considering that the latter implementation allows for faster retrieval of prediction results, we use the latter one in our experiments.

**Architecture and Settings.** We use the same setup for all baselines and our method. Specifically, following previous worksWang et al. (2020); Li et al. (2022); Zhang et al. (2021a), we employ ResNet-32 for CIFAR100/10-LT, ResNeXt-50/ResNet-50 for ImageNet-LT, ResNet-152 for Places-LT and ResNet-50 for iNaturalist 2018 as backbones, respectively. If not specified, we use the SGD optimizer with a momentum of 0.9 and set the initial learning rate to 0.1 with linear decay.

**Others.** The results of the comparison methods are taken from their original paper, and our results are averaged over three experiments. In experiments combining our method with other long-tail algorithms, we adopt the same optimal hyper-parameters reported in their original papers. More implementation details and the hyper-parameter statistics are reported in the Appendix A.3.

### 4.2 COMPARISONS WITH SOTA ON BENCHMARKS.

**Baselines.** As a general consistency regularization method to deal with the bias of tail classes, the proposed IKL can be integrated into prevalent LT methods. We followed previous works Zhang et al. (2021b) to summarize the investigated LT algorithms into three categories: 1) one- or two-stage rebalancing, 2) augmentation, and 3) ensemble learning methods.

For re-balancing approaches, we studied two-stage re-sampling methods cRT Kang et al. (2019) and LWS Kang et al. (2019), multi-branch models with diverse sampling strategies like BBN Zhou et al. (2020), and reweight loss functions like Balanced Softmax (BSCE) Ren et al. (2020); Menon et al. (2020a) and LDAM Cao et al. (2019).

For augmentation approaches, we empirically noticed that some common data augmentation methods are more general and effective than other long-tailed transfer learning methods, so we adopted Random Augmentation (RandAug) Cubuk et al. (2020) in our experiments.

For ensemble learning methods, we followed the recent trend of ensemble learning like NCL Li et al. (2022), SADE Zhang et al. (2022) and RIDE Wang et al. (2020), which are proved to be state-of-the-art models in LT classification that are capable of improving both head and tail categories at the same time. In particular, we also compare with the SOTA method based on multi-expert knowledge distillation Li et al. (2022).

| Method | CIFAR-100-LT | | |
|---|---|---|---|
| IF | 10 | 50 | 100 |
| Softmax | 59.1 | 45.6 | 41.4 |
| BBN | 59.8 | 49.3 | 44.7 |
| BSCE | 61.0 | 50.9 | 46.1 |
| RIDE | 61.8 | 51.7 | 48.0 |
| SADE | 63.6 | 53.9 | 49.4 |
| Softmax+IKL | 59.6(+0.5) | 46.2(+0.4) | 41.9(+0.5) |
| BSCE+IKL | 64.5(+3.5) | 52.2(+1.3) | 47.9(+1.8) |
| RIDE+IKL | 62.4(+0.6) | 53.1(+1.4) | 48.8(+0.8) |
| SADE+IKL | 64.5(+0.9) | 55.4(+1.5) | 50.7(+1.3) |
| BSCE† | 63.0 | - | 50.3 |
| PaCo† | 64.2 | 56.0 | 52.0 |
| SADE† | 65.3 | 57.3 | 53.2 |
| BSCE+IKL† | 64.6(+1.6) | - | 51.2(+0.9) |
| PaCo+IKL† | 65.1(+0.9) | 57.1(+1.1) | 52.8(+0.8) |
| SADE+IKL† | 66.8(+1.5) | 59.1(+1.4) | 54.7(+1.5) |

Table 1: Comparisons on CIFAR100-LT datasets with the IF of 10, 50 and 100. †denotes models trained with RandAugmentCubuk et al. (2020) for 400 epochs.

| Method | Many | Medium | Few | All |
|---|---|---|---|---|
| Softmax | 68.1 | 41.5 | 14.0 | 48.0 |
| Decouple-LWS | 61.8 | 47.6 | 30.9 | 50.8 |
| BSCE | 64.1 | 48.2 | 33.4 | 52.3 |
| LADE | 64.4 | 47.7 | 34.3 | 52.3 |
| PaCo | 63.2 | 51.6 | 39.2 | 54.4 |
| RIDE | 68.0 | 52.9 | 35.1 | 56.3 |
| SADE | 66.5 | 57.0 | 43.5 | 58.8 |
| Softmax+IKL | 68.6(+0.5) | 42.0(+0.5) | 14.7(+0.7) | 48.6(+0.6) |
| BSCE+IKL | 65.6(+1.5) | 49.7(+1.5) | 37.9(+4.5) | 54.8(+2.5) |
| PaCo+IKL | 64.0(+0.8) | 52.5(+0.9) | 42.1(+2.9) | 56.4(+2.0) |
| RIDE+IKL | 68.9(+0.9) | 54.1(+1.2) | 38.6(+3.5) | 59.0(+2.7) |
| SADE+IKL | 66.3(-0.2) | 58.3(+1.3) | 47.8(+4.3) | 60.2(+1.4) |
| PaCo† | 67.5 | 56.9 | 36.7 | 58.2 |
| SADE† | 67.3 | 60.4 | 46.4 | 61.2 |
| PaCo+IKL † | 67.4(-0.1) | 57.3(+0.4) | 37.8(+1.1) | 58.8(+0.6) |
| SADE+IKL † | 67.9(+0.7) | 61.2(+0.8) | 47.8(+1.4) | 62.0(+0.8) |

Table 2: Comparisons on ImageNet-LT. † denotes models trained with RandAugmentCubuk et al. (2020) for 400 epochs.

**Superiority on Long-tailed Benchmarks.** This subsection compares IKL with state-of-the-art long-tailed methods on vanilla long-tailed recognition. Table 1, 2, 3, and 4 lists the Top-1 accuracy of SOTA methods on CIFAR-100-LT, ImageNet-LT, Places-LT, and iNaturalist 2018, respectively. Our approach seamlessly integrates with existing methods, yielding performance improvements across all long-tail benchmarks. Notably, when applied to the SADE method on the ImageNet-LT dataset, our approach achieves a maximum performance boost of 4.3% in few-shot. In Appendix, IKL also outperforms baselines in experiments on long-tail CIFAR-10.

| Method | Many | Medium | Few | All |
|---|---|---|---|---|
| Softmax | 46.2 | 27.5 | 12.7 | 31.4 |
| BLS | 42.6 | 39.8 | 32.7 | 39.4 |
| LADE | 42.6 | 39.4 | 32.3 | 39.2 |
| RIDE | 43.1 | 41.0 | 33.0 | 40.3 |
| SADE | 40.4 | 43.2 | 36.8 | 40.9 |
| Softmax+IKL | 46.1(-0.1) | 28.0(+0.3) | 15.6(+2.9) | 32.8(+1.4) |
| BLS+IKL | 43.0(+0.4) | 40.3(+0.5) | 34.8(+2.1) | 41.1(+1.7) |
| LADE+IKL | 42.8(+0.2) | 39.7(+0.3) | 35.5(+3.2) | 41.8(+2.6) |
| RIDE+IKL | 43.1(+0.0) | 41.9(+0.9) | 36.9(+3.9) | 42.1(+1.8) |
| SADE+IKL | 41.0(+0.6) | 44.3(+1.1) | 38.7(+1.9) | 42.2(+1.1) |
| PaCo † | 36.1 | 47.2 | 33.9 | 41.2 |
| NCL | - | - | - | 41.5 |
| PaCo+IKL † | 36.4(+0.3) | 47.7(+0.5) | 43.9(+2.7) | 42.8(+1.6) |

Table 3: Comparisons on Places-LT, starting from an ImageNet pre-trained ResNet-152 provided by Torchvision. †denotes models trained with RandAugmentCubuk et al. (2020) for 400 epochs.

| Method | Many | Medium | Few | All |
|---|---|---|---|---|
| Softmax | 74.7 | 66.3 | 60.0 | 64.7 |
| BLS | 70.9 | 70.7 | 70.4 | 70.6 |
| LADE† | 64.4 | 47.7 | 34.3 | 52.3 |
| MiSLAS | 71.7 | 71.5 | 69.7 | 70.7 |
| RIDE | 71.5 | 70.0 | 71.6 | 71.8 |
| SADE | 74.5 | 72.5 | 73.0 | 72.9 |
| Softmax+IKL | 75.4(+0.7) | 67.1(+0.8) | 61.1(+1.1) | 65.5(+0.8) |
| BLS+IKL | 68.8(+2.1) | 72.5(+1.8) | 75.9(+5.5) | 73.1(+2.5) |
| LADE+IKL | 64.8(+0.4) | 48.9(+1.1) | 36.6(+2.3) | 73.6(+1.8) |
| RIDE+IKL | 71.4(+0.1) | 70.9(+0.9) | 74.8(+3.2) | 73.6(+1.8) |
| SADE+IKL | 74.7(+0.2) | 73.1(+0.6) | 77.8(+4.8) | 74.2(+1.3) |
| PaCo † | 69.5 | 73.4 | 73.0 | 73.0 |
| SADE † | 75.5 | 73.7 | 75.1 | 74.5 |
| NCL | 72.7 | 75.6 | 74.5 | 74.9 |
| PaCo+IKL † | 69.6(+0.1) | 73.4(+0.0) | 75.9(+1.9) | 73.6(+0.6) |
| SADE+IKL † | 75.7(+0.2) | 74.1(+0.4) | 77.8(+2.7) | 75.3(+0.8) |
| NCL+IKL † | 72.5(-0.2) | 76.7(+1.1) | 77.8(+3.3) | 76.5(+1.6) |

Table 4: Comparisons on iNaturalist 2018. † denotes models trained with RandAugmentCubuk et al. (2020) for 400 epochs.

**IKL with different backbone results.** Table 2 shows that ECL obtains consistent performance improvements on various backbones. Whether the backbone is CNN-based networks (ResNet, ResNext) or Transformer-based networks (Swin Tiny and Small), IKL delivers consistent accuracy gains.

| Method | Resnet-50 | ResNeXt-50 | Swin-T | Swin-S |
|---|---|---|---|---|
| Softmax | 41.6 | 44.4 | 42.6 | 42.9 |
| OLTR | - | 46.3 | - | - |
| $\tau$-norm | 46.7 | 49.4 | - | - |
| cRT | 47.7 | 49.9 | - | - |
| LWS | 47.3 | 49.6 | - | - |
| LDAM | - | - | 50.6 | 49.5 |
| RIDE | 54.9 | 56.4 | 56.3 | 54.2 |
| Softmax+IKL | 45.8(+4.2) | 47.3(+3.1) | 43.7(+1.1) | 43.6(+0.7) |
| $\tau$-norm | 47.3(+0.6) | 50.5(+1.1) | - | - |
| cRT | 48.5(+0.8) | 51.2(+1.3) | - | - |
| LWS | 48.5(+1.2) | 50.5(+0.9) | - | - |
| LDAM+IKL | - | - | 52.1(+1.5) | 50.3(+0.8) |
| RIDE+IKL | 56.8(+1.9) | 58.7(+2.3) | 59.1(+2.8) | 55.6(+1.4) |

Table 5: Comparisons on ImageNet-LT with different backbones.

| Method | Many | Med | Few | All |
|---|---|---|---|---|
| Softmax | 66.1 | 37.3 | 10.6 | 41.4 |
| OLTR | 61.8 | 41.4 | 17.6 | - |
| $\tau$-norm | 65.7 | 43.6 | 17.3 | 43.2 |
| cRT | 64.0 | 44.8 | 18.1 | 43.3 |
| LDAM | 61.5 | 41.7 | 20.2 | 42.0 |
| RIDE | 69.3 | 49.3 | 26.0 | 48.0 |
| SADE | 60.3 | 50.2 | 33.7 | 49.4 |
| Softmax+IKL | 66.8(+0.7) | 37.9(+0.6) | 11.2(+0.6) | 41.9(+0.5) |
| LDAM+IKL | 62.4(+0.9) | 42.4(+0.7) | 28.3(+2.3) | 49.2(+1.2) |
| RIDE+IKL | 69.9(+0.6) | 50.4(+1.1) | 28.1(+2.1) | 49.2(+1.2) |
| SADE+IKL | 60.4(+0.1) | 50.8(+0.6) | 35.5(+1.8) | 50.7(+1.3) |

Table 6: Comparisons on CIFAR-100-LT(IF=100) with different sample sizes.

**IKL contributes to different sample size results.** To explore the reasons why IKL works for long-tail scenarios, we provide a more detailed and comprehensive evaluation. Specifically, we divide the classes into multiple categories based on their sample size, namely, Many (with more than 100 images), Medium (with 20 to 100 images), and Few (with less than 20 images). Softmax trains the model with only cross-entropy, so it simulates the long-tailed training distribution and performs well on many-shot classes. However, it performs poorly on medium-shot and few-shot classes, leading to worse overall performance. In contrast, re-balanced long-tailed methods (e.g., Decouple, Causal) seek to simulate the uniform class distribution for better average performance, but they inevitably sacrifice the performance on many-shot classes. Table 2, 4 and 6 demonstrates the significant enhancement in the performance of few- and medium-shot classes achieved by the proposed IKL, while maintaining high accuracy for many-shot classes. Moreover, there is a slight improvement observed in the performance of many-shot classes.

## 5 COMPONENT ANALYSIS AND ABLATION STUDY

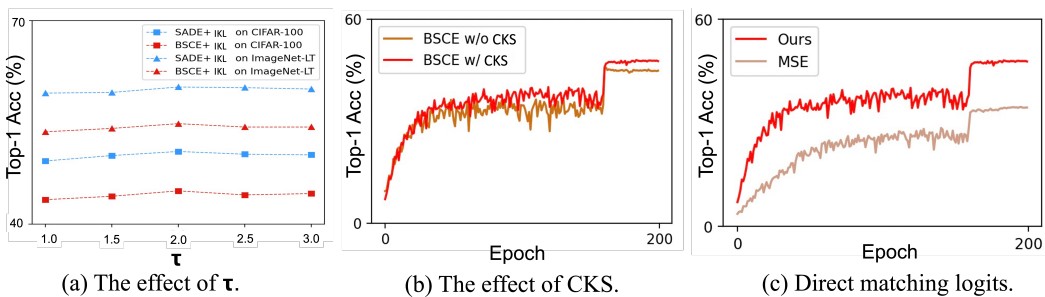

(a) The effect of $\tau$.      (b) The effect of CKS.      (c) Direct matching logits.

Figure 4: Figure (a): The effect of temperature $\tau$ for different methods and datasets. Figure (b): The effect of our CKS. Figure (c): The effect of directing matching logits.

**The effective of our components IUR and ICL.** Our proposed method is fundamentally composed of two primary components: Implicit Uncertainty regularization (IUR) and Implicit Correlation Labeling (ICL). As shown in Tab 7, the IUR component is designed to enforce consistency across all categories. As a result, it notably enhances the accuracy of the tail classes, but this comes at the expense of a slight reduction in the accuracy of the head classes. In contrast, ICL facilitates learning across all categories by leveraging the inherent feature correlations, compensating for the minor drawbacks introduced by IUR and ensuring an overall improved performance.

**The effective of temperature $\tau$.** The temperature parameter $\tau$ is introduced to soften the previous predictions, allowing the current model to learn from a smoother, more generalized distribution. By adjusting the temperature parameter during training, we can control the trade-off between accuracy and generalization to optimize the current prediction. Higher temperature values lead to better generalization but lower accuracy, while lower temperature values lead to better accuracy but less

generalization. In Figure. 4 (a), we show several settings of $\tau$ on the CIFAR-100LT (IF=100) and ImageNet-LT, we observe that when the $\tau$ set to 2, the models achieve the best performance.

| IUR | ICL | Many | Med | Few | All |
|---|---|---|---|---|---|
| - | - | 66.1 | 37.3 | 10.6 | 41.4 |
| - | ✓ | 66.6 | 37.5 | 10.9 | 41.6 |
| ✓ | - | 65.5 | 37.7 | **11.4** | 41.5 |
| ✓ | ✓ | **66.8** | **37.9** | 11.2 | **41.9** |

Table 7: **Ablation study on the components of our methods.** Comparisons on the baseline model (Softmax) for CIFAR-100-LT(IF=100) with different component combinations.

| Strategy | $\alpha$ | Top-1 Acc. |
|---|---|---|
| Equal weight | 0.5 | 64.7 |
| Linear Increment | $T/T_{max}$ | 65.8 |
| Cosine Increment | $[1 - \cos(\pi \cdot T/T_{max})]/2$ | 66.5 |
| Learnable Parameter | - | 65.2 |
| Parabolic Increment | $(T/T_{max})^2$ | **66.8** |

Table 8: **Ablation studies of different progressive factor strategies.** Comparisons on the baseline model (Softmax) for CIFAR-100-LT (IF=100) with different progressive factor strategies $\alpha$.

**The effect of $\alpha$.** The value of $\alpha$ critically dictates the effectiveness of our proposed method. It modulates the weightage of $\mathcal{L}_{IKL}$ throughout the training process. From the Tab 8, we can observe that the strategy employed to adjust $\alpha$ impacts the model's Top-1 accuracy. Specifically, when employing the Cosine Increment and Parabolic Increment strategies, our method exhibits enhancements over the conventional softmax baseline (66.1%). Conversely, strategies such as Equal weight, Linear Increment, and Learnable Parameter induce varying degrees of performance degradation relative to the softmax baseline. This decrement can be attributed to the suboptimal feature representations in the early training phase, leading to inferior soft label generation by both IUR and ICL. Therefore, a high $\alpha$ value in the preliminary training stages can introduce adverse effects, given the potential propagation of erroneous soft labels.

**The effect of our CKS.** The component CKS also plays a key role in the training process. During the learning process, the CKS filters out the probability distribution of incorrect predictions from the output of the previous epoch. It ensures the distribution of our current prediction to avoid wrong information. In Figure 4 (c), we show top-1 test accuracy of BSCE+IKL w/ our CKS and BSCE+IKL w/o our CKS on CIFAR-100LT (IF=100). The results demonstrate that our IKL with CKS leads to a significant improvement.

**Direct matching logits.** There is another approach in IUR to regularizing the uncertainty, such as using Mean Square Error (MSE) to direct matching logits. The object function denotes:

$$\mathcal{L}_{MSE} = \frac{1}{2}(v_{i,t-1} - v_{i,t})^2 \tag{10}$$

If we are in the high-temperature limit, our IUR is equivalent to minimizing Eq. 10, provided the logits are zero-meaned separately for each transfer case Hinton et al. (2015). In Figure 4, we visualize the test accuracy based on BSCE with $\mathcal{L}_{MSE}$ on CIFAR-100LT (IF=100). However, we observe it has a rapid decline in results compared with our IUR. Because at lower temperatures, IUR pays much less attention to matching logits that are much more negative than the average. This has the potential advantage that these logits are almost completely unconstrained by the cost function used to train the model, so they can be very noisy Hinton et al. (2015).

## 6 CONCLUSION

In this paper, we propose Implicit Knowledge Learning (IKL), is a plug-and-play method for improving long-tailed recognition (LTR) in computer vision. It contains Implicit Uncertainty Regularization (IUR) and Implicit Correlation Labeling (ICL) and addresses two key challenges in the long-tail methods: (1) the implicit uncertainty during training process (2) a marked bias in the model's prediction towards the head class. Experimental results on popular benchmarks demonstrate the effectiveness of our approach, consistently outperforming state-of-the-art methods by 1% to 5%. IKL seamlessly integrates with existing LTR methods and is compatible with various backbone architectures, making it a practical and versatile solution for improving LTR performance.

**Limitation.** For our IKL, the predictions from the model at (t-1)-th epoch are necessary for training at t-th epoch. When working with large datasets, such as tens of thousands of categories, this can lead to additional memory consumption.

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

# A    APPENDIX

## A.1    MORE ANALYSIS OF IMPLICIT KNOWLEDGE LEARNING.

### A.1.1    OUR IKL BOOSTS LTR MODELS LEARNING MORE KNOWLEDGE, ESPECIALLY FOR RARE CLASSES.

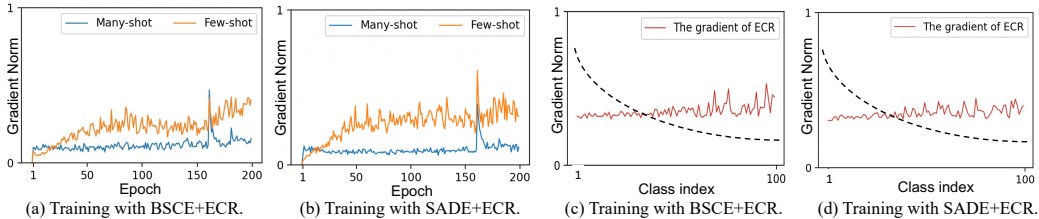

(a) Training with BSCE+ECR.    (b) Training with SADE+ECR.    (c) Training with BSCE+ECR.    (d) Training with SADE+ECR.

Figure 5: We conduct these experiments based on BSCE Ren et al. (2020) and SADE Zhang et al. (2022) on CIFAR-100LT (IF=100). Figure (a) and (b): The trend of the gradient in many- and few-shot training with BSCE+IKL and SADE+IKL. Figure (c) and (d): The gradient for all classes at the end epoch (T) training with BSCE+IKL and SADE+IKL.

To make more clear why the proposed IKL boosts the existing LTR methods, we first explore this question by analysis the gradient of $\mathcal{L}_{IKL}$.

In Figure 5 (a) and (b), we visualize the ground-truth class' gradient of $\mathcal{L}_{IKL}$ of the many-shot (with more than 100 images) classes and few-shot (with less than 20 images) classes on CIFAR-100LT (IF=100) during the training process. Then, in Figure 5 (c) and (d), we visualize the gradient of $\mathcal{L}_{IKL}$ across all classes at the end training epoch. As shown in Figure 5 (a) and (b), the contributed gradient of $\mathcal{L}_{IKL}$ of the few-shot classes gradually increase. In contrast, the many-shot classes contribute fewer gradients. The gradient varies widely at the 160-th epoch because the learning rate is decaying at this point. Furthermore, as indicated in Figure 5 (c) and (d) we show the gradient across all classes at a random epoch, we also observe the tail classes contribute more gradients for the training model. These observations demonstrate that gradients produced by regularizing the current predictions and learning form the feature Correlation label may provide more knowledge than supervised by one-hot labels.

### A.1.2    OUR IKL BOOSTS LTR MODELS LEARNING BETTER CLASSIFIER.

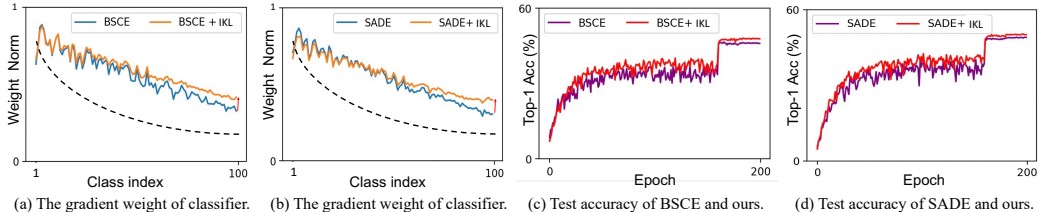

(a) The gradient weight of classifier.    (b) The gradient weight of classifier.    (c) Test accuracy of BSCE and ours.    (d) Test accuracy of SADE and ours.

Figure 6: We conduct these experiments based on BSCE Ren et al. (2020) and SADE Zhang et al. (2022) on CIFAR-100LT (IF=100). Figure (a) and (b): The average L2 norm of the gradient of weights in the classifier. Figure (c): The test top-1 accuracy for BSCE and BSCE+IKL (ours). Figure (c): The test top-1 accuracy for SADE and SADE+IKL (ours).

To understand the transferred knowledge from our IKL that boosts the classifier, we also visualize the average L2 norm of the gradient of weights in the classifier in Figure 6 (a) and (b). As indicated in Figure 6 (a) and (b), the LTR methods' classifier with IKL present more balanced classified gradient

weights, especially in rare classes. Moreover, as indicated in Figure 6 (c) and (d), the LTR methods with our IKL achieve better test performance.

## A.2   DATASET

**CIFAR-100/10-LT.** CIFAR-100/10-LT is a variant of CIFAR-100/10 Krizhevsky et al. (2009) dataset with imbalanced class distributions. The original CIFAR-100/10 dataset comprises 50,000 training images and 10,000 validation images of size $32 \times 32$, and consists of 100/10 classes. To ensure a fair comparison, we follow the same long-tailed version as used in the previous works Wang et al. (2020); Zhang et al. (2021a). In our experiments, we use the Imbalance Factor (IF) $\beta$ to quantify the degree of class imbalance in the data, which is defined as $\beta = N_{max}/N_{min}$. In our experiments, we evaluate our approach on CIFAR-100-LT datasets with imbalance factors set to 100, 50, and 10, respectively. The results of CIFAR-10-LT are reported in Appendix.

**ImageNet-LT** and **Places-LT**. ImageNet-LT and Places-LT are the long-tailed variants of the popular ImageNet-2012 Deng et al. (2009) and the large-scale scene classification dataset Places Zhou et al. (2017), respectively. Liu et al. Liu et al. (2019) proposed these datasets and we follow their work for a comparable evaluation. We sample subsets of the datasets following the Pareto distribution with the power value $\gamma = 6$, as done in their work. ImageNet-LT comprises 115.8K images belonging to 1000 categories, with an imbalance factor of $\beta = 1280/5$. Similarly, Places-LT contains 184.5K images from 365 categories, with an imbalance factor of $\beta = 4980/5$.

**iNaturalist 2018**. iNaturalist Horn et al. (2017) is a large-scale real-world dataset specifically designed for long-tailed visual recognition, characterized by an extremely imbalanced class distribution. The dataset consists of 437.5K training images and 24.4K validation images belonging to 8142 categories. The presence of fine-grained categories further exacerbates the already challenging task of recognition Wei et al. (2019).

## A.3   MORE IMPLEMENTATION DETAILS OF OUR METHOD

We implement our method in PyTorch. FollowingWang et al. (2020); Zhang et al. (2022), we use ResNeXt-50 for ImageNet-LT, ResNet-32 for CIFAR10/100-LT, ResNet-152 for Places-LT and ResNet-50 for iNaturalist 2018 as backbones, respectively.

More detailed statistics of network architectures and hyper-parameters are reported in Table9. Based on these hyper-parameters, we conduct experiments on 1 TITAN RTX 2080 GPU for CIFAR100-LT, 4 GPUs for iNaturalist 2018, and 2 GPUs for ImageNet-LT and Places-LT, respectively.

In the training phase, the data augmentations are the same as previous long-tailed studies Kang et al. (2019); Zhang et al. (2022). If not specified, we use the SGD optimizer with a momentum of 0.9 and set the initial learning rate as 0.1 with linear decay. More specifically, for ImageNet-LT, we train models for 180 epochs with batch size 64 and a learning rate of 0.025 (cosine decay). For CIFAR100-LT, the training epoch is 200 and the batch size is 128. For Places-LT, followingLiu et al. (2019), we use ImageNet pre-trained ResNet-152 as the backbone, while the batch size is set to 128 and the training epoch is 30. Besides, the learning rate is 0.01 for the classifier and 0.001 for all other layers. For iNaturalist 2018, we set the training epoch to 200, the batch size to 512, and the learning rate to 0.2. For all datasets, we set the temperature $\tau$ to 2.

| Items | CIFAR100/10-LT | ImageNet-LT | Places-LT | iNaturalist 2018 |
|---|---|---|---|---|
| Network Architectures | | | | |
| network backbone | ResNet-32 | ResNeXt-50 | ResNet-152 | ResNet-50 |
| Training Phase | | | | |
| epochs | 200/400 | 180/400 | 30 | 100/400 |
| batch size | 64 | 256 | 64 | 512 |
| learning rate (lr) | 0.1 | 0.1 | 0.01 | 0.2 |
| lr schedule | linear decay | cosine decay | linear decay | linear decay |
| temperature $\tau$ | 2 | 2 | 2 | 2 |
| weight decay factor | $5 * 10^{-4}$ | $5 * 10^{-4}$ | $5 * 10^{-4}$ | $5 * 10^{-4}$ |
| momentum factor | 0.9 | | | |
| optimizer | SGD optimizer with nesterov | | | |

Table 9: Statistics of the used network architectures and hyper-parameters in our experiments.

