# OpenReview forum: "IKL: Boosting Long-Tail Recognition with Implicit Knowledge Learning"
_ICLR.cc/2024/Conference — ICLR 2024 Conference Withdrawn Submission_

### Official Review · Reviewer_W2Ce · 2023-10-27

**Soundness:** 2 fair
**Presentation:** 3 good
**Contribution:** 2 fair
**Rating:** 3
**Confidence:** 5

**Summary:**

This paper focuses on addressing the long-tailed problem from the perspective of the training process uncertainty and model prediction correlation. it proposes an  Implicit Knowledge Learning method which consists of an Implicit Uncertainty Regularization (IUR) for mimicking the prediction behavior over adjacent epochs and an Implicit Correlation Labeling (ICL) to reduce the bias introduced by one-hot labels. Experiments are conducted on various long-tailed dataset.

**Strengths:**

1. The proposed idea is clear and easy to understand.
2. The authors commit to open-sourcing the code to facilitate result reproduction.
3. The authors discuss some potential limitations, such as computational costs.
4. The IKL is a plug-and-play method which could be plugged into many existing long-tailed solutions and bringing performance improvement.

**Weaknesses:**

1. While some improvement can be observed in the tail classes, currently, there is no concrete evidence to support the claim that learning between two adjacent epochs can enhance the model's performance. The authors should provide more theoretical justification for this claim rather than relying solely on empirical observations from training experiments.
2. IKL appears to be a regularization-based approach to network learning, and indeed, there are other regularization-based solutions for long-tail problems (e.g., WD[1]). It would be beneficial for the authors to provide a more detailed comparison between IKL and these existing solutions.
3. In Table 7, IUR and ICL don't seem to have brought significant improvements to the results, especially given that this is on a smaller dataset CIFAR-100-LT. This raises concerns about the performance of IKL. The authors should address this concern by providing a more in-depth explanation or conducting additional experiments to demonstrate the effectiveness of IKL.

[1] Alshammari, Shaden, et al. "Long-tailed recognition via weight balancing." Proceedings of the IEEE/CVF Conference on Computer Vision and Pattern Recognition. 2022.

**Questions:**

1. The authors propose IKL as a solution to address the issue of model prediction uncertainty. In fact, many expert methods like RIDE and SADE are also designed based on the same principle. The authors should provide a detailed explanation of why IKL results in greater improvements when combined with these expert methods compared to using Softmax (e.g., results in Table5, combined with Softmax only improves the performance by 0.5, but combined with RIDE improves 1.4).

---

> ### Author Response · Authors · 2023-11-22
> **Response to Reviewer W2Ce (Part 1)**
>
> We appreciate your thoughtful review of our work. We address your questions below:
>
> ```
> W1: Concrete evidence to support the claim that learning between two adjacent epochs can enhance the model's performance.
> ```
>
> Thank you for your insightful feedback. We explain why learning between two adjacent epochs can enhance the model's performance from the following viewpoints:
>
> **1. Why does uncertainty of predictions exist during training on long-tail data?**
>
> Firstly, due to the application of dropout and data augmentation regularization techniques, the network output during training becomes a stochastic variable [1].
> Particularly under a long-tail distribution, we frequent use of strong data augmentation methods like RandAugment [2] significantly contributes to this variability. Such augmentations, including complex transformations like Gaussian filtering, induce more intricate changes in the input $x_i$, thereby markedly increasing the uncertainty in the model’s output for these samples.
> Therefore, two evaluations of the same input $x_i$ can yield uncertain results even under the same network weights $\theta$.
>
> Furthermore, training models often exhibit underfitting for tail classes [3]. This phenomenon of underfitting further exacerbates the prediction uncertainty for these tail-class samples as shown in Fig 1.
>
> In summary, it is the combination of these effects that explain the observed uncertainty/inconsistency between the prediction vectors $v_i$ and $\hat{v}_i$ from two adjacent epochs, with this being particularly pronounced in tail classes."
>
> **2. Why guarantee consistency for the predictions to improve the performance of recognition?**
>
> When a percept is changed slightly, a human typically still considers it to be the same object. Correspondingly, a classification model should favor functions that give consistent output for similar data points [4].
> Therefore, aiming to minimize this difference to be consistent is a reasonable goal.
>
> Specifically, consistency regularization works by reducing the cost loss associated with the manifold surrounding each data point [4].
> Finally, guaranteeing consistency for the predictions effectively moves the decision boundaries further from the points with labels and improves the performance of recognition.
>
> [1] Samuli Laine et al., Temporal Ensembling for Semi-Supervised Learning. ICLR, 2017.
>
> [2] Ekin D Cubuk et al., Randaugment: Practical automated data augmentation with a reduced search space. CVPR, 2020.
>
> [3] Zhang, Yifan, et al. "Deep long-tailed learning: A survey." TPAMI, 2023.
>
> [4] Mean teachers are better role models: Weight-averaged consistency targets improve semi-supervised deep learning results, NIPS 2017.

---

> ### Author Response · Authors · 2023-11-22
> **Response to Reviewer W2Ce (Part 2)**
>
> ```
> W2: Compare our ICL with other regularization-based solutions methods.
> ```
>
> Below, we provide a detailed discussion and comparison with other regularization-based methods such as Weight Balance [1]. We appreciate this opportunity to address this issue.
>
> Additional experiments are conducted to evaluate and integrate our method with regularization-based methods such as Mixup [2], Weight Balance [1], and MiSLAS [3]. The Mixup [2] stands as a representative method for data augmentation regularization, enhancing model generalization by interpolating between samples. The Weight Balance [1] directly constrains the weights from the classifier through a regularization term, addressing the imbalance by modulating the impact of more frequent classes. The MiSLAS [3] introduces label-aware smoothing as a regularization strategy, aimed at mitigating varying degrees of over-confidence across different classes. Unlike these methods above, our method designs a regularization loss to reduce the uncertainty of the predictions during training and provide class correlation labels for boosting existing long-tailed methods.
>
> In our experiments, it is important to note that:
>
> - We selected the state-of-the-art combination of **WD + WD & Max** from the original paper on Weight Balance [1] for comparison across all datasets used in that study.
> - Both MiSLAS and Weight Balance, the two regularization methods designed for long-tail distribution, employ a decoupled two-stage training approach. Therefore:
> a) We compared these methods with a baseline decoupled training method designed for long-tail distribution [4], termed as **Decouple**.
> b) For a fair comparison, we also combined the decoupled training approach with IKL (Decouple + IKL), to compare it against MiSLAS and Weight Balance methods.
> c) For the Mixup results presented in the tables, we also utilized a decoupled training implementation.
>
>
> | Method           | CIFAR100-LT | ImageNet-LT | iNaturalist 2018 |
> |------------------|-------------|-------------|------------------|
> | Decouple          |     43.8        |    47.9          |       67.7           |
> | Mixup             |     45.1        |    51.5          |       70.0           |
> | MiSLAS            |     47.0        |      52.7        |           71.6       |
> | WD + WD & Max     |     53.6        |      53.9        |           70.2       |
> | Decouple + **IKL**   |  50.9        |      54.5        |            72.8      |
> | MiSLAS & **IKL**     |  53.1        |      56.0        |            74.2      |
> | WD & **IKL** + WD & Max & **IKL**          |     56.8        |      56.7        |           73.5       |
>
> Table 1. Results of comparing and combining our method with other regularization-based methods.
>
> The table above illustrates that our method outperforms other regularization-based methods under a decoupled two-stage training setting.
> Additionally, the integration of other regularization-based methods into our method results in further enhancements to performance.
> This improvement substantiates the orthogonality and potential synergistic relationship between our approach and other regularization-based methods.
>
> [1] Alshammari, Shaden, et al. Long-tailed recognition via weight balancing. CVPR, 2022.
>
> [2] HongyiZhang, et al. mixup: Beyond empirical risk minimization. ICLR, 2018.
>
> [3] Zhisheng Zhong, et al. Improving Calibration for Long-Tailed Recognition. CVPR, 2021.
>
> [4] Bingyi Kang, et al. Decoupling representation and classifier for long-tailed recognition. ICLR, 2020.

---

> ### Author Response · Authors · 2023-11-22
> **Response to Reviewer W2Ce (Part 3)**
>
> ```
> W3: provide a more in-depth explanation or conduct additional experiments.
> ```
>
> Thank you for your valuable suggestion and the opportunity to clarify our approach.
>
> We will address the effectiveness of our method from the following aspects:
>
> **1). Explanation of results on cross-entropy loss (Softmax)**
>
> In Table 7 of our paper, the improvement for IUR and ICL modules are 0.2% and 0.3%, respectively. The reason for this is that the ablation experiment results were obtained by training the model using cross-entropy loss.
> Specifically, when training with Cross-entropy loss on long-tailed datasets, the model exhibits a clear prediction bias towards the head classes, meaning it is more likely to misclassify tail classes as head classes, and the prediction confidence for head class samples is higher.
>
> In this case, for IUR, there is a higher consistency of predictions in head class predictions between adjacent epochs, and our confidence knowledge selection (CKS) module in IUR could only select a small number of correctly predicted tail class samples and filter out most of the error prediction. In other words, only a few correctly predicted tail class samples contribute to the IUR loss. Thus, the contribution of the IUR loss improves the model performance by 0.2%.
>
> For ICL, since the model is trained using cross-entropy loss without any re-weighting methods, its classifier is imbalanced, and the similarity matrix calculated through the features also contains significant bias. Thus, the contribution of the ICL loss improves the model performance by 0.3%.
>
> These issues are in the simple baseline Cross-Entropy loss, and our method is very effective for the long-tailed methods. We analyze in the next part **Q1**: Why IKL performs better on long-tailed methods than on cross-entropy loss. Next, we present the results of our IUR and ICL are effective in boosting the existing long-tailed methods.
>
> **2). Additional Ablation Experiments**
>
> Therefore, we provide ablation experiments on all four datasets with multiple methods. For reasons why IKL performs better on these methods than on Cross-Entropy, please refer to the next **Q1**.
>
> As shown in the following table, using ICL or IUR with RIDE results in a maximum improvement of 2.1%, and 1.7% across the three datasets. Furthermore, using ICL or IUR with SADE results in a maximum improvement of 1.2%, and 1.1% across the three datasets. Results show significant improvement when using ICL and IUR separately, which demonstrates the effectiveness of both designs.
>
> |      |      | CIFAR100-LT |  CIFAR100-LT | ImageNet-LT |  ImageNet-LT   | iNaturalist 2018 |  iNaturalist 2018     |
> |------|------|-------------|---------|-------------|---------|------------------|---------|
> | IUR  | ICL  | RIDE        | SADE    | RIDE        | SADE    | RIDE             | SADE    |
> | -    | -    | 48.0        | 49.4    | 56.3        | 58.8    | 71.8             | 72.9    |
> | -    | ✅   | 48.6        | 50.1    | 58.4        | 60.0    | 72.7             | 73.6    |
> | ✅   | -    | 48.6        | 49.9    | 58.0        | 59.7    | 73.2             | 74.0    |
> | ✅   | ✅   | 48.8        | 50.7    | 59.0        | 60.2    | 73.6             | 74.2    |
>
>
> ```
> Q1: Explanation of why IKL results in greater improvements when combined with these expert methods compared to using Softmax.
> ```
>
> Thank you for your helpful suggestion. We explain this reason below:
>
> **1. For Multi-Expert Methods** (e.g. SADE and RIDE)
>
> When applying multi-expert methods like SADE and RIDE, these approaches improve the performance of the architecture, reducing the prediction uncertainty for head classes and increasing correct predictions for more tail class samples.
> With more correct tail-class predictions, our CKS module (in IUR) could select more correctly predicted tail-class samples that exhibit inconsistency in adjacent predictions. This also means that using IUR, there are more correctly predicted tail-class samples which are with uncertainty predictions. These more samples would contribute more to the loss and lead to a more significant improvement (e.g., 1.4% with RIDE).
>
> **2. For Softmax Baseline**
>
> Without any reweighting/ensemble methods (i.e., Softmax), the model tends to be biased on predictions of head class, misclassifying most tail class samples during training. Since our approach CKS selects only correctly predicted samples for loss contribution, the contribution from tail classes is less (as analyzed in **W3**), resulting in a weak improvement compared to RIDE and SADE.
>
> In general, long-tail methods like SADE exhibit greater uncertainty for tail classes, with the inconsistency arising from more correct predictions of tail-class samples. In contrast, models trained with Softmax exhibit incorrect predictions of tail class, which contain less information and are then filtered out by CKS and also do not contribute to the loss.
>
> Overall, we sincerely appreciate all your time and valuable feedback!

---

> ### Author Response · Authors · 2023-11-23
> **Looking forward to further discussion**
>
> Dear Reviewer W2Ce,
>
> We genuinely appreciate any feedback you may have on our revised submission. If there are additional questions or points of clarification needed, we are more than willing to promptly address them.
>
> Again, we express our gratitude for your time, extensive efforts, and valuable insights. Thank you for the opportunity to engage in this discussion.
>
> Best regards, The Authors

---

### Official Review · Reviewer_Zu8m · 2023-10-30

**Soundness:** 2 fair
**Presentation:** 3 good
**Contribution:** 2 fair
**Rating:** 5
**Confidence:** 5

**Summary:**

1. This paper aims to address long-tailed recognition via knowledge distillation.

2. Considering the prediction uncertainty of models trained in adjacent epochs, the authors propose to use the model trained in the last epoch to guide the training in the current epoch.

3. Inspired by this idea, an $L_{IUR}$ loss is proposed by directly distilling knowledge with KL divergence loss from the model trained in the last epoch.

4. Moreover,  models trained by cross-entropy suffer from classifier bias. The paper proposes to use medium-feature to construct a new classifier. Based on the new classifier, it distills knowledge from the model trained in the last epoch again.

**Strengths:**

1. The paper is clear and easy to follow.
2. The method is simple but effective. Improvements are observed when combining it with previous methods.

**Weaknesses:**

1. The proposed L_{IUR} loss uses KL loss to regularize the output from the current model to be similar to the output from the model of the last epoch.
    Because the last epoch model is fixed, the KL loss is actually equal to a cross-entropy loss with soft labels from the last epoch model.
    The proposed L_{ICL} loss uses the pseudo-label from the last epoch model. The pseudo-label is calculated based on the medium-feature classifier.
    The difference between L_{IUR} and L_{ICL} is that L_{IUR} distills knowledge with a biased classifier while L_{ICL} distills knowledge with a medium-feature classifier without bias.
    Thus, from my point of view, the proposed method is a kind of ensemble of distillation with a rebalanced classifier (with L_{ICL}) and a biased classifier trained with cross-entropy (with L_{IUR}).

2. The authors claim that uncertainty of models trained in adjacent epochs exists. Is it really important?
    If we use only one well-trained teacher model through the training, it can also help models give consistent predictions.
    The paper should show the necessity of distillation with the model trained in the last epoch rather than a specific well-trained teacher model.

3. Comparison with previous distillation-based methods are missed.

4. What's the medium function in Eq. (7)? How to rank the features and find the medium?

**Questions:**

See weakness.

**Details Of Ethics Concerns:**

No ethics concerns.

---

> ### Author Response · Authors · 2023-11-22
> **Response to Reviewer  Zu8m (Part 1)**
>
> Thank you for your constructive feedback. We address your concerns below:
>
> ```
> W1&W2: The meaning and importance of consistency regularization.
> ```
>
> Thank you for your valuable feedback. We explain the meaning and importance of consistency regularization from the following viewpoints: Why does inconsistency or uncertainty of predictions exist during training on long-tail data?; and 2) Why guarantee consistency for the predictions to improve the performance of recognition?
>
> **1). Why does the uncertainty of predictions exist during training on long-tail data?**
>
> Firstly, due to the application of dropout and data augmentation regularization techniques, the network output during training becomes a stochastic variable [1]. Particularly under a long-tail distribution, our frequent use of strong data augmentation methods like RandAugment [2] significantly contributes to this variability. Such augmentations, including complex transformations like Gaussian filtering, induce more intricate changes in the input, thereby markedly increasing the uncertainty in the model’s output for these samples. Therefore, two evaluations of the same input could yield uncertain results even under the same network weights.
>
> Furthermore, training models often exhibit underfitting for tail classes [3]. This phenomenon of underfitting further exacerbates the prediction uncertainty for these tail-class samples as shown in Fig 1.
>
> In summary, it is the combination of these effects that explains the observed uncertainty/inconsistency between the prediction vectors and from two adjacent epochs, with this being particularly pronounced in tail classes.
>
> **2). Why guarantee consistency for the predictions to improve the performance of recognition?**
>
> When a percept is changed slightly, a human typically still considers it to be the same object. Correspondingly, a classification model should favor functions that give consistent output for similar data points [4]. Therefore, aiming to minimize this difference to be consistent is a reasonable goal.
>
> Specifically, consistency regularization works by reducing the cost loss associated with the manifold surrounding each data point [4]. Finally, guaranteeing consistency for the predictions effectively moves the decision boundaries further from the points with labels and improves the performance of recognition.
>
> [1] Samuli Laine et al., Temporal Ensembling for Semi-Supervised Learning. ICLR, 2017.
>
> [2] Ekin D Cubuk et al., Randaugment: Practical automated data augmentation with a reduced search space. CVPR, 2020.
>
> [3] Zhang, Yifan, et al. "Deep long-tailed learning: A survey." TPAMI, 2023.
>
> [4] Mean teachers are better role models: Weight-averaged consistency targets improve semi-supervised deep learning results, NIPS 2017.
>
> ```
> W2: Comparison with a well-trained teacher.
> ```
>
> To conduct a fair comparison between the use of a well-trained teacher and the predictions from past epochs as the teacher, we maintain all other settings constant but replace the self-distillation component of IKL with distillation using a well-trained teacher.
>
> Taking SADE as an example, we first train a model utilizing the SADE method to establish a well-trained teacher model. Then, we train a new model using the SADE with the IKL method. During this new model's training phase, we replace IKL’s self-distillation mechanism with distillation from the well-trained teacher (SADE + IKL w/ well-trained teacher).
>
> Our results are summarized in the following table. We conduct experiments on CIFAR100-LT, ImageNet-LT, and iNaturalist.
>
> |   Method    | CIFAR100-LT | ImageNet-LT | iNaturalist |
> |-------------|-------------|-------------|-------------|
> |   RIDE [1] + IKL w/ well-trained teacher         |    48.3         |     57.9        |    72.3         |
> |   RIDE [1] + IKL             |    48.8         |     59.0        |    73.6         |
> |   SADE [2] + IKL w/ well-trained teacher         |    49.0         |     59.1        |    73.3         |
> |   SADE [2] + IKL             |    50.7         |     60.2        |    74.2         |
>
>
> The results indicate that compared to traditional distillation methods using a well-trained teacher, our approach demonstrates more significant improvements in model performance. Additionally, our method does not require a well-trained model as a teacher, thereby substantially reducing additional computational overhead and resource burdens.
>
> [1]  Wang X, Lian L, Miao Z, et al. Long-tailed Recognition by Routing Diverse Distribution-Aware Experts. ICLR, 2020.
>
> [2] Yifan Zhang, et al. Zhang Self-supervised aggregation of diverse experts for test-agnostic long-tailed recognition. NIPS, 2022

---

> ### Author Response · Authors · 2023-11-22
> **Response to Reviewer Zu8m (Part 2)**
>
> ```
> W3: Comparison with previous distillation-based methods are missed.
> ```
>
> We compare our method IKL with the following previous distillation-based methods: LFME [1] and NCL [2].
>
> LFME employs self-paced knowledge distillation from multiple models to enhance learning on long-tailed data, while NCL introduces Nested Balanced Online Distillation (NBOD) for effective knowledge transfer among multiple expert networks to tackle long-tailed recognition challenges.
>
> It is noteworthy that, to conduct a comprehensive comparison with these distillation-based methods, we not only substituted the distillation approaches of LFME and NCL with our own self-distillation method, denoted by an arrow (e.g., NCL->IKL), but we also augmented their distillation frameworks with our self-distillation approach, indicated by a plus sign (e.g., NCL+IKL).
>
>
> |   Method    | CIFAR100-LT | ImageNet-LT | iNaturalist |
> |-------------|-------------|-------------|-------------|
> |   LFME                    |    42.3         |     37.2        |    -         |
> |   LFME -> IKL             |    44.0         |     37.9        |    -         |
> |   LFME + IKL              |    44.8         |     38.9        |    -         |
> |   NCL                    |    54.2         |     59.5        |    74.9         |
> |   NCL -> IKL             |    54.5         |     60.2        |    75.2         |
> |   NCL + IKL              |    56.0         |     61.9        |    76.5         |
>
>
> The results in the table above demonstrate that replacing the distillation component of these frameworks from LFME and NCL with our self-distillation method leads to improved model performance.
> Furthermore, when our self-distillation method is combined with these methods, the model performance can be further enhanced.
>
> **Reference**
>
> [1] Xiang L, Ding G, Han J. Learning from multiple experts: Self-paced knowledge distillation for long-tailed classification[C]//Computer Vision–ECCV 2020: 16th European Conference, Glasgow, UK, August 23–28, 2020, Proceedings, Part V 16. Springer International Publishing, 2020: 247-263.
>
> [2] Li J, Tan Z, Wan J, et al. Nested collaborative learning for long-tailed visual recognition[C]//Proceedings of the IEEE/CVF Conference on Computer Vision and Pattern Recognition. 2022: 6949-6958.
>
> ```
> W4: the medium function in Eq. (7).
> ```
>
> We provide an explanation of how to compute the medium function in Eq. (7):
>
> Given a class $C$ with $n$ samples and $m$ features, represented as a matrix $X$ of size $n \times m$ and $x_{ij}$ denotes the value of the i-th sample in the j-th feature dimension.
>
> #### Steps to Calculate the class prototype $\mathcal{f_c}$:
>
> 1. **Sorting**: For each feature dimension $j$ (where $j = 1, 2, \ldots, m$), sort all sample values $x_{ij}$ in ascending order.
>
> 2. **Median Calculation**:
>    - If the number of samples $n$ is **odd**, the median for each feature dimension $j$ is the middle value in the sorted list.
>    - If $n$ is **even**, the median for each feature dimension $j$ is the average of the two middle values.
>
> 3. **Class Prototype Formation**:
>    - The class prototype $\mathcal{f_c}$ is a vector of medians for each feature dimension, represented as:
>    $\mathcal{f_c}$ = ($C_1$, $C_2$, $\ldots$, $C_m$)
>    - Where $C_j$ is the median of the $j$-th feature dimension.
>
> In code, we could use the median function in Pytorch to easily implement this procedure.
>
> In conclusion, we sincerely appreciate all your valuable feedback to help us improve our work.

---

> ### Author Response · Authors · 2023-11-23
> **Looking forward to further discussion**
>
> Dear Reviewer Zu8m,
>
> We genuinely appreciate any feedback you may have on our revised submission. If there are additional questions or points of clarification needed, we are more than willing to promptly address them.
>
> Again, we express our gratitude for your time, extensive efforts, and valuable insights. Thank you for the opportunity to engage in this discussion.
>
> Best regards, The Authors

---

### Official Review · Reviewer_6AeQ · 2023-10-30

**Soundness:** 3 good
**Presentation:** 3 good
**Contribution:** 2 fair
**Rating:** 6
**Confidence:** 4

**Summary:**

This paper proposes a framework named Implicit Knowledge Learning (IKL) framework to tackle the long-tailed recognition problem. In detail, the IKL framework includes two techniques, so called the Implicit Uncertainty Regularization (IUR) and the Implicit Correlation Labeling (ICL). First, the main idea of IKL is to regularize the predictions of the current epoch using the ones from the previous epoch. It is especially shown to be effective to reduce uncertainty in the tail-class examples. Second, ICL constructs an additional label matrix based on the inter-class similarity, to improve the learning process. This can help to complement the typical one-hot labels. The full framework IKL can serve as a plug-and-play scheme, where it can be attached to existing long-tailed learning methods.

**Strengths:**

- The necessity of IUR technique is quite clear and noticeable, as presented in Figure 1. Such discovery, where uncertainty values especially on the minor classes grow compared to the ones from the major categories is useful. The proposed scheme can properly address the problem. In addition, learning from correlations between different classes is reasonable.

- The proposed framework is practical, since it can be built on existing methods to further improve long-tailed learning.

- Experiments are extensive; it has been tested on more than three or four different types of baseline methods, while demonstrating consistent improvements.

**Weaknesses:**

- The critical downside of the proposed work is about technical novelty. The proposed IKL combines two components IUR and ICL, based on using previous predictions and inter-class correlations respectively, both of which are related to well-established literature. For example, as the elementary deep learning based semi-supervised learning methods, Temporal Ensembling [S. Laine et al., 2016] and MeanTeacher [A. Tarvainen et al., 2017] present the generalized version of IUR, where averaging past model predictions or model parameters to enforce consistency loss term. In that sense, despite the limited novelty, the proposed work needs to discuss the previous works in this direction and experimentally compare with those methods in the long-tailed learning scenario. Similarly, regarding the proposed ICL technique, it is common to learn from the dependencies among different class labels (also in the name of co-occurrence) [Z. M. Chen et al., 2019]. Authors also need to acknowledge previous attempts in this direction and provide sufficient discussions on what component is new. In summary, from the technical aspect, it provides limited innovation compared to previous literature that adopts similar approaches.

- As an additional comment on ICL, the effect for applying ICL is currently unclear. To better understand the proposed component, it would be beneficial to quantitatively measure and qualitatively visualize the class-level dependencies (i.e., correlation matrix) on a specific dataset.

- For calculating the class prototypes C, the verification for the superiority of median features compared to simple averaging is missing. Is there any reference or supporting experiments? In addition, it would be further helpful to provide a brief explanation about how to compute a median of features.

- A proof-reading process, especially for referencing papers, is necessary. A lot of typos can be found, for example, missing a space (i.e., NCLLi et al.) and missing punctuation (i.e., offline process Peng et al.). Additionally, the reference section needs to be updated. For example, the paper ‘Balanced meta-softmax…’ from Jiawei Ren et al., is presented in NeurIPS 2020, which is written as arXiv in the current version.

**Questions:**

- To further improve reproducibility, it is recommended to provide the set of parameters for each augmentation operation in RandAugment.

- Related to the limitation provided in the conclusion, specific demonstration for the increase of space or memory by saving the previous epoch’s predictions and computing class-wise similarity matrix, depending on the number of total samples and classes, would be helpful for better understanding this limitation.

- It needs to mention that the progressive scaling of $\alpha$ is applied from section 3 for better clarity. It is confusing since it firstly appears in the experiment section.

---

[Summary and guidance to rebuttal]

Overall, the motivation for the proposed IUR and ICL is quite clear in the context of long-tailed learning and it conveys consistent improvements in experiments. However, as those techniques are not totally new, it is necessary to present connections with previous approaches in those directions and to emphasize the novel aspects of the proposed work.

---

References

[S. Laine et al., 2016] Temporal ensembling for semi-supervised learning, in ICLR 2016.

[A. Tarvainen et al., 2017] Mean teachers are better role models: Weight-averaged consistency targets improve semi-supervised deep learning results, in NIPS 2017.

[Z. M. Chen et al., 2019] Multi-Label Image Recognition with Graph Convolutional Networks, in CVPR 2019.

---

Update: The rebuttal partially addressed my concerns, which are presenting connections to previous literature and additional experiments.

---

> ### Author Response · Authors · 2023-11-22
> **Response to Reviewer 6AeQ (Part 1)**
>
> We sincerely appreciate your valuable feedback on our paper and thank you for taking the time to review it. Below, we outline our rebuttal, addressing the specific points you mentioned:
>
> ### W1.1 Compare our IUR with the semi-supervised methods, such as Temporal Ensembling and MeanTeacher.
>
> To clarify the novelty design of our IUR with Temporal Ensembling and MeanTeacher, we illustrate from two aspects: the details of methods, and additional experiments:
>
> 1. The details of our IUR.
>
> To apply a consistency cost between the two predictions and better leverage the implicit knowledge of the long-tailed labeled data, we design the confidence knowledge selection (CKS) module and select the predictions from adjacent epochs.
>
> - The CKS module:
>
> With long-tailed **labeled** data, we design the confidence knowledge selection (CKS) module for the consistency process to leverage these labels, ensuring the accuracy of the prediction of the last epoch. In contrast, Temporal Ensembling [1] and MeanTeacher [2] maintain an exponential moving average (EMA) for past model predictions or model parameters, without leveraging the power of labels which may produce inaccurate predictions. If too much weight is given to inaccurate predictions, the cost of inconsistency outweighs that of misclassification, preventing the learning of new information.
>
> - Selecting predictions from adjacent epochs:
>
> Due to a classification model that should favor functions that give consistent output for similar data points [2], our IUR achieves this by reducing the uncertainty of the predictions from one model at two epochs (e.g., t1 and t2) with twice-augmented inputs and this uncertainty is amplified in long-tail data. To ensure that the uncertainty of predictions stems from the input, we utilize the training model which is at minimum variation (e.g., at adjacent epochs). If we utilize EMA, the knowledge gained in the early training would be involved. Due to insufficient training, this knowledge may contain biased knowledge.
>
>
> 3. The Experimental Results:
>
> |    Top-1 Acc.     | ImageNet-LT | iNaturalist 2018 |
> |-------------------|----------|------------------|
> | BSCE [3] | 52.3 | 70.6 |
> | + Temporal Ensembling | 51.5 | 69.4 |
> | + MeanTeacher         | 51.8 | 70.1 |
> | + IUR w/o CKS         | 51.7 | 70.2 |
> | + IUR (Ours)          | 54.2 | 72.3 |
>
> We have also compared our method to semi-supervised consistency prediction methods on ImageNet-LT and iNaturalist 2018. The results demonstrate that, due to Temporal Ensembling and MeanTeacher being designed for unlabeled data, they perform poorly in a long-tailed setting. In contrast, due to the design of CKS and selecting predictions from adjacent epochs, our approach significantly outperforms these semi-supervised methods in long-tailed recognition.
>
> In summary, we design several novels for IUR to leverage the knowledge of **labeled** long-tail data. In this way, we ensure the consistency of the model's prediction uncertainty in long-tailed data and facilitate the model to learn more abstract invariance [1] to improve the performance of recognition. In the final vision, we will add these related methods to make our approach clearer.
>
> **Reference**
>
> [1] S. Laine et al., 2016, Temporal ensembling for semi-supervised learning, in ICLR 2016.
>
> [2] A. Tarvainen et al., 2017 Mean teachers are better role models: Weight-averaged consistency targets improve semi-supervised deep learning results, in NIPS 2017.
>
> [3] Ren J, Yu C, Ma X, et al. Balanced meta-softmax for long-tailed visual recognition[J]. Advances in neural information processing systems, 2020, 33: 4175-4186.

---

> ### Author Response · Authors · 2023-11-22
> **Response to Reviewer 6AeQ (Part 2)**
>
> ### W1.2 Compare our ICL with co-occurrence methods.
>
> To illustrate the difference between our ICL and co-occurrence methods, we first review the motivation of our ICL, we then introduce the difference from the definition perspective and methodological perspective.
>
> 1. The motivation of our ICL.
>
> Traditional one-hot labels lack the ability to convey information about inter-class similarities. In long-tail learning scenarios, head classes might exhibit features similar to those of tail classes. Relying on one-hot encoding can lead to misclassification of these tail class features as belonging to head classes, exacerbating the model's bias towards head classes. To this end, our Implicit Correlation Labeling (ICL) addresses this issue by introducing correlation labels, enabling the model to learn both the associations and distinctions between classes, thereby mitigating model bias.
>
> We differentiate our method from the co-occurrence approach [1] from two perspectives:
>
> 2. Definition perspective:
>
> Our proposed method fundamentally diverges from co-occurrence-based approaches. The "co-occurrence" [1] is designed to model **inter-class dependencies** for *multi-label classification* tasks. For instance, 'bird' is likely to co-occur with 'sky' or 'tree'. In multi-label scenarios, these co-occurring relationships among labels serve as priors, assisting in the recognition of complex patterns and enhancing classification performance.
>
> However, in long-tail recognition tasks that focus on single-label classification, the knowledge of inter-class dependency (co-occurrence) is irrelevant. Our ICL approach is specifically developed for *long-tail tasks*, creating an **inter-class feature similarity matrix**. For example, 'cat' is more similar to 'tiger' than to 'dog' in terms of features. This similarity matrix mixes the labels to supervise single categories, ensuring the model reduces misclassified features of tail classes as belonging to head classes due to their similarity, thereby reducing model bias towards these predominant classes.
>
> 3. Methodological perspective:
>
> The co-occurrence method [1] initially utilizes a Graph Convolutional Network (GCN) to model each category's dependencies, combining this dependency vector with standard prediction results for multi-label classification. Conversely, our ICL method directly leverages the features from the previous epochs to compute an inter-class similarity matrix. This matrix is then used as a soft label to supervise the model.
>
> In summary, compared with the co-occurrence method [1], our ICL method does not introduce additional networks to model inter-class relationships. Instead, it employs the relationship matrix directly as a supervisory signal, simplifying the process and providing direct supervision based on these similarity relationships.
>
> **Reference**
>
> [1] Z. M. Chen et al., 2019, Multi-Label Image Recognition with Graph Convolutional Networks, in CVPR 2019.

---

> ### Author Response · Authors · 2023-11-22
> **Response to Reviewer 6AeQ (Part 3)**
>
> ### W3: For calculating the class prototype C, the verification for the superiority of median features compared to simple averaging is missing.
>
> When the model training on the long-tailed dataset, especially large-scale datasets, the previous utilized rand augmentation to increase model generalization. The rand augmentation contains strong augmentation (e.g. color distortion and jigsaw transformation) and may augment an input to be a noisy point [1][2]. So, we obtain the class centers with median values instead of the class mean to avoid the impact of noisy points.
>
> In addition, we provide a brief explanation of how to compute a median of features:
>
> Given a class $C$ with $n$ samples and $m$ features, represented as a matrix $X$ of size $n \times m$ and $x_{ij}$ denotes the value of the i-th sample in the j-th feature dimension.
>
> #### Steps to calculate the class prototype $\mathcal{f_c}$:
>
> 1. **Sorting**: For each feature dimension $j$ (where $j = 1, 2, \ldots, m$), sort all sample values $x_{ij}$ in ascending order.
>
> 2. **Median Calculation**:
>    - If the number of samples $n$ is **odd**, the median for each feature dimension $j$ is the middle value in the sorted list.
>    - If $n$ is **even**, the median for each feature dimension $j$ is the average of the two middle values.
>
> 3. **Class Prototype Formation**:
>    - The class prototype $\mathcal{f_c}$ is a vector of medians for each feature dimension, represented as:
>    $\mathcal{f_c}$ = ($C_1$, $C_2$, $\ldots$, $C_m$)
>    - Where $C_j$ is the median of the $j$-th feature dimension.
>
> In code, we could use the median function in Pytorch to easily implement this procedure. In the below table, we compare the median feature and mean feature for the class prototype.
>
> |    Top-1 Acc.     | ImageNet-LT | iNaturalist 2018 |
> |-------------------|------------------|------------------|
> | RIDE+Mean  | 57.7 |72.1 |
> | RIDE+Median (Ours) |   58.4 | 72.7 |
> | SADE+Mean  |   59.3 | 73.2 |
> | SADE+Median (Ours) |   60.0 | 73.6 |
>
> The results demonstrate the effectiveness of the median feature.
>
> **Reference**
>
> [1] Peng X, Wang K, Zhu Z, et al. Crafting better contrastive views for siamese representation learning[C]//Proceedings of the IEEE/CVF Conference on Computer Vision and Pattern Recognition. 2022: 16031-16040.
>
> [2] Lee H, Lee K, Lee K, et al. Improving transferability of representations via augmentation-aware self-supervision[J]. Advances in Neural Information Processing Systems, 2021, 34: 17710-17722.
>
> [3] Ren J, Yu C, Ma X, et al. Balanced meta-softmax for long-tailed visual recognition[J]. Advances in neural information processing systems, 2020, 33: 4175-4186.
>
> ### W4: Typos.
>
> Thank you for pointing out the typographical and referencing errors. We will follow all your suggested revisions and conduct a comprehensive review of the entire paper to ensure its accuracy and quality in the final version.
>
> ### Q1: Provide Details in RandAugment.
>
> Thank you for your recommendation to enhance the reproducibility of our work.
> For a fair comparison, our parameters follow PaCo [1], which has demonstrated effectiveness in similar contexts.
>
> Specifically, we set the RandAugment parameters as follows in our experiments:
>
> - Magnitude (m): 10
> - Number of Transformations (n): 3
> - Weight Index (w): 0
> - Magnitude Standard Deviation (mstd): 0.5
>
> **Reference**
>
> [1] Cui, Jiequan, et al. "Parametric contrastive learning." Proceedings of the IEEE/CVF international conference on computer vision. 2021.
>
> ### Q2: Resource consumption.
>
> In the below table, we outline the resource consumption required due to storing predictions for each epoch and the memory required for the class-wise similarity matrix. We present the data in a table format for the long-tail datasets: CIFAR100-LT, ImageNet-LT, Place-LT, and iNaturalist 2018.
>
>
> |                   | ImageNet-LT | Place-LT | iNaturalist 2018 |
> |-------------------|-------------|----------|------------------|
> | Memory for Previous Epoch  | 0.045 GB | 0.089 GB | 1.31 GB |
> | Memory for Similarity Matrix | 0.004 GB | 0.005 GB | 0.12 GB |
>
> The above table illustrates the memory requirements for the Previous Epoch Storage and the Similarity Matrix across different datasets. For datasets of normal scale, such as ImageNet-LT and Place-LT, the combined memory usage of both methods does not exceed 1GB, which is generally manageable. However, for larger datasets, like iNaturalist 2018, the Previous Epoch Storage method alone can require significant memory resources, approximately 1.31 GB in this case.
>
> Therefore, the limitation in terms of storage space is primarily evident in large-scale datasets. For small to medium-sized datasets, these methods do not require much additional memory. This distinction highlights the need for careful consideration of memory resources when applying our methodology to large datasets.

---

> ### Author Response · Authors · 2023-11-22
> **Response to Reviewer 6AeQ (Part 4)**
>
> ### Q3: The progressive scaling of $\alpha$ is applied from section 3 for better clarity.
>
> In the initial stages of training, due to the model's limited capability from insufficient training, it is particularly challenging to extract valuable implicit knowledge. Consequently, at this early training phase, the losses derived from the model's previous outputs — which serve to reduce uncertainty ($L_{IUR}$) and compute class centers ($L_{ICL}$) — are significantly biased and meaningless. Therefore, their contribution to the overall training process needs to be diminished to prevent the propagation of early inaccuracies. As training progresses and the model becomes more capable, the knowledge extracted becomes increasingly reliable and valuable. In response to this, we wish to increment their contribution, allowing the model to leverage more of this valuable knowledge.
>
> In light of this, we designed a progressive scaling weight $\alpha$ to ensure minimal impact on normal training at the outset while increasing influence during the later stages, enabling the model to assimilate more valuable knowledge. The application of $\alpha$ is detailed in Section 3.4:
>
> $$ L = (1 - \alpha)L_{LTR} + \alpha(L_{IUR} + L_{ICL}) $$
>
> For the selection of $\alpha$, our intent is for it to grow progressively from the start to the end of training. Based on the experiments detailed in Section 5, we adopted the Parabolic Increment strategy for $\alpha$'s growth. Specifically:
>
> $$ \alpha = (T / T_{max})^2 $$
>
> This approach, as confirmed by our experiments, allows the model to gradually integrate knowledge from previous epochs, effectively balancing the training influence over time.
>
> Overall, we appreciate all your constructive feedback and the opportunity to improve our work.

---

> ### Comment · Reviewer_6AeQ · 2023-11-23
> **Response to authors' comments**
>
> Thank you for your comprehensive feedback, which includes detailed clarifications, discussions, and follow-up experiments. I have carefully checked the discussions with other reviewers as well.
>
> **W1.1 Compare our IUR with the semi-supervised methods, such as Temporal Ensembling and MeanTeacher.**
>
> I appreciate your effort in clarifying the IUR method and providing experimental comparisons with Temporal Ensembling and MeanTeacher. However, IUR appears to be a specific instance of consistency regularization, a well-established concept in semi-supervised learning (SSL). Additionally, the experimental results do not demonstrate a significant improvement of IUR over MeanTeacher. I am also curious about the performance drop observed when consistency regularization terms are applied to BSCE without CKS. While BSCE is not the primary focus in the main paper's experimental section, additional validation, possibly on smaller-scale benchmarks like CIFAR-10/100 using different LT-baseline methods such as RIDE and SADE, would be valuable. Furthermore, the CKS module's concept seems derivative of pseudo-labeling [D. H. Lee, 2013], with the primary distinction being that CKS uses given labels instead of confidence-based filtering. Therefore, the technical novelty and effectiveness of the IUR method remain unclear at current phase.
>
> **W1.2 Compare our ICL with co-occurrence methods.**
>
> Thank you for the thorough discussion on the co-occurrence-based method. Upon revisiting the ICL section, I find that leveraging inter-class similarity based on prototypes for loss computation is already explored [O. Atsuro, 2022]. Could you elucidate any novel aspects or differences from existing literature?
>
> **W3: For calculating the class prototype C, the verification for the superiority of median features compared to simple averaging is missing.**
>
> Your clarification and supportive experiments are much appreciated! Including this in the appendix could be beneficial.
>
> ---
>
> References
>
>
> [D. H. Lee, 2013] "Pseudo-label: The simple and efficient semi-supervised learning method for deep neural networks." Workshop on challenges in representation learning, ICML. Vol. 3. No. 2. 2013.
>
> [O. Atsuro, 2022] "Interclass prototype relation for few-shot segmentation." European Conference on Computer Vision. Cham: Springer Nature Switzerland, 2022.
>
> ---
>
> In summary, while I value the extensive responses from the authors, there are still unresolved questions regarding the technical contributions.

---

> > ### Author Response · Authors · 2023-11-23
> > **Response to Reviewer 6AeQ (Part 1)**
> >
> > Thank you for your insightful viewpoints and timely discussions with us.
> >
> >
> > ### Q1 The technical novelty of our IUR and with the MeanTeacher and the pseudo-labeling method.
> >
> > Our IUR uses the given labels to filter out the predictions. This means that only samples with correct predictions participate in consistency regularization. When the prediction of the sample is incorrect but the confidence of the prediction is high, our method could still filter this sample. In contrast, the pseudo-labeling method may not be able to filter this sample because its filtering condition is **without class label information, just a confidence threshold**. Also, the MeanTeacher fails to **filter the tail class predictions** that are incorrectly predicted, resulting in the current sample learning biased consistency information.
> >
> >
> > ### Q2 The effectiveness of our IUR.
> >
> > Below, we conduct more validation of our effectiveness based on RIDE and SADE on the CIFAR-10LT and CIFAR-100LT datasets. The imbalance factor of both datasets is 100. For the Pseudo-labeling, we replace the CKS with pseudo-labeling in the IUR and set the filter threshold to 0.8 for comparison.
> >
> >
> > |         | CIFAR10-LT |  CIFAR10-LT | CIFAR100-LT |  CIFAR100-LT |
> > |------ |-------------|---------|-------------|---------|
> > | IUR     | RIDE      | SADE    | RIDE        | SADE    |
> > | Baseline |    81.6             | 83.8          |  48.0        | 49.4    |
> > | MeanTeacher [1] | 81.2 (-0.4)      | 83.3 (-0.5)   |  47.8 (-0.2) | 49.0 (-0.4)    |
> > | IUR w/o CKS | 81.4 (-0.2)      | 83.4 (-0.4)   |  47.6 (-0.4) | 49.1 (-0.3)    |
> > | IUR w/ Pseudo-labeling [2] | 81.9 (+0.3)      | 83.9 (+0.1)   |  48.2 (+0.2) | 49.6 (+0.2)    |
> > | IUR (Our) | 82.3 (+0.7)     | 84.4  (+0.6)  | 48.6  (+0.6)      | 49.9 (+0.5)   |
> >
> > (1) Comparison with the MeanTeacher method [1].
> >
> > When we employ MeanTeacher in the long-tailed method, it causes a performance drop for the method. **This is due to the unfiltered incorrect predictions used for consistency regularization and the accumulation of historically incorrect predictions, where the model learns biased information from unfiltered knowledge.** In contrast, we design the CKS to select the correct prediction, helping the model to ensure consistency and improve identification.
> >
> > (2) Comparison with the Pseudo-labeling method [2].
> >
> > We observe that IUR with pseudo-labeling has a weak improvement compared to CKS, which is more effective, mainly due to the fact that our CKS prediction filtering condition takes into account the prediction **as the correct category in addition to the confidence thresholds.**
> >
> > (3) The performance drops without CKS.
> >
> > In a long-tailed distribution, the model tends to predict samples as head categories and underfit tail categories. **This tail class underfitting phenomenon leads to some incorrect predictions of the tail class by the model during training.** If we perform consistency regularization without CKS when the prediction of the samples in the previous epoch is incorrect, this would lead to the samples in the current epoch learning from incorrect knowledge. This process leads to a drop in recognition effectiveness, so CKS is necessary to help filter out incorrect knowledge.
> >
> > In addition, for the IUR, we also have insight contributions. Training on a long-tailed dataset, the model has a prediction preference for the head class and underfits the tail class, so it amplifies the prediction uncertainty of the augmented inputs as shown in Fig. 1. Therefore, for long-tail methods, **it is necessary to ensure prediction consistency**, and we have made a preliminary exploration of the methods without changing the pipeline and significantly improved the effectiveness of the **existing** long-tail methods.
> >
> > **Reference**
> >
> > [1] S. Laine et al., 2016, Temporal ensembling for semi-supervised learning, in ICLR 2016.
> >
> > [2] D. H. Lee, 2013, "Pseudo-label: The simple and efficient semi-supervised learning method for deep neural networks." Workshop on challenges in representation learning, ICML. Vol. 3. No. 2. 2013.

---

> > ### Author Response · Authors · 2023-11-23
> > **Response to Reviewer 6AeQ (Part 2)**
> >
> > ### Q3 Elucidate novel aspects or differences from existing literature.
> >
> > In our ICL we calculate the prototypes from **the median features of all samples** for each class. Our approach is easily **plug-and-played** into any of the existing long-tail methods. In contrast, the few-shot segmentation method, IPRM [1], generates the prototype of **mean features in a mini-batch** for each class. For long-tailed data, the major method is the uniform sample method. **This means that in a batch, the head class is in the majority and the tail class rarely occurs.** Therefore, there is a large bias for sampling only one batch in the long-tailed data to compute the category prototypes, and some tailed categories may not contain samples. Although we may mitigate this problem by balancing sampling a batch of data or including more batches, this limits the **portability** of the approach.
> > In addition, the IPRM [1] calculates the prototype using the **mean method**, and in the answer to **W3**, we also showed the advantage of using **median method**: the reduction of noise point interference improves the recognition (about **0.5%–1%**). In conclusion, our method is specifically designed to compute class prototypes in long-tail scenarios and is superior to IPRM in both of the above points.
> >
> > **Reference**
> >
> > [1] O. Atsuro, 2022, "Interclass prototype relation for few-shot segmentation." European Conference on Computer Vision. Cham: Springer Nature Switzerland, 2022.
> >
> > ### Q4: Your clarification and supportive experiments are much appreciated! Including this in the appendix could be beneficial.
> >
> > Thank you for your constructive feedback. We will include this in the appendix of the final version.
> >
> > In conclusion, our technical innovations, although somewhat similar to previous methods, are carefully designed in a long-tail scenario and are effective when applied to existing long-tail methods. Besides, the insight of our viewpoint, including that **long-tailed data amplifies the prediction uncertainty of tailed samples in training** and that **one-hot labeling exacerbates the bias of the model towards the header class**, is an important contribution to the long-tailed community. We sincerely appreciate all your time and valuable feedback to help us improve our work. We hope that you reconsider our submission with the revised version.

---

> ### Comment · Reviewer_6AeQ · 2023-11-23
> **Response to the authors' comments**
>
> Thank you for your prompt and thorough feedback to my response!
>
> Based on the comments you presented, while simple confidence-based filtering, as an alternative to CKS, can be beneficial, CKS appears to have a more positive effect in the long-tailed learning scenario. To validate this hypothesis, it would be useful to apply CKS to other consistency regularization methods such as MeanTeacher and Temporal Ensembling. As the author-reviewer discussion period is approaching its end, please consider addressing this in the post-review revision phase. In this context, it would be better to mention the observation in the manuscript that the CKS technique is an essential component.
>
> I also appreciate your making a conceptual contrast with the above previous work (IPRM).
>
> In summary, the proposed work could become more comprehensive and unique if the aspects discussed above (i.e., connections to semi-supervised learning, the effect of CKS compared to confidence-based filtering, and conceptual contrast to the co-occurrence-based method and the similar prototype-based method IPRM) are incorporated into the revised draft.
>
> If all these aspects are addressed, I would be inclined to recommend a positive score (i.e., 5 -> 6).

---

> > ### Author Response · Authors · 2023-11-23
> > **Thank you for considering raising the score.**
> >
> > Thank you for considering raising the score. In the post-review revision phase, we plan to verify the effectiveness of CKS by applying it to other consistency regularization methods, such as MeanTeacher and Temporal Ensembling. We will also incorporate the above discussions into the revised draft to underline the uniqueness of our work. We truly appreciate your thorough review and have found the discussions with you both insightful and enriching!

---

### Meta-Review · Area_Chair_peEw · 2023-12-07

**Metareview:**

The paper proposes to conduct Implicit Knowledge Learning (IKL) that extracts the hidden knowledge in long-tail learning processes, aiming to improve performance in long-tail recognition. The proposed IKL consists of two key components: Implicit Uncertainty
Regularization (IUR) and Implicit Correlation Labeling (ICL), where the former considers the uncertainty of the predictions over adjacent epochs and the latter leverages  inter-class similarity information. Experiments conducted on commonly used benchmark datasets demonstrate the effectiveness of the proposed approach.

A major concern shared by the reviewers is whether the proposed IKL has sufficient technical novelty as the two key components have some major overlaps with existing methods. As reviewers pointed out, there have already been existing efforts using the last epoch checkpoint as a teacher model to distill knowledge for the current epoch training. Regularization-based solutions have also been used for long-tail problems. Additionally, to make a convincing case, more concrete evidence (e.g., theoretical analysis) should be provided to justify why the proposed design can benefit long-tailed recognition.

**Justification For Why Not Higher Score:**

The overall technical novelty does not appear to be significant and important (theoretical) evidence is missing to adequately justify the effectiveness of the proposed approach for long-tailed recognition.

**Justification For Why Not Lower Score:**

N/A

---

### Decision · Program_Chairs · 2024-01-16

Reject